# Position: We Need Practical AI Alignment Methods to Mirror Human Reasoning

Vijay Keswani [1]  Breanna K. Nguyen [2]  Cyrus Cousins [2]
Vincent Conitzer [3]  Walter Sinnott-Armstrong [2]  Jana Schaich Borg [4]

## Abstract

AI systems are increasingly employed as decision aids, decision delegates, or autonomous decision-makers. This position paper argues that in many settings, particularly high-stakes decision-making, we need accurate *cognitively-aligned* AI systems that *reason similarly to their users*, and *faithfully communicate* their reasoning. We review evidence that cognitive alignment improves understandability and trustworthiness, and provide new survey data showing that many users find cognitive alignment "essential" when an AI's rationale for a judgment or action is important to them. We outline the gaps between existing alignment methods and what is needed to achieve cognitive alignment, and present a research agenda to address these gaps. We argue that cognitive misalignment represents a likely impediment to AI adoption in many envisioned applications, and that addressing it is important for creating AI systems on which users are both willing and justified to rely.

## 1. Introduction

If you were a literary editor considering using AI to help make initial content reviews, would you prefer (**A**) an AI that uses your personal editorial standards and style preferences, or (**B**) an AI that produces similar judgments to (**A**), but via opaque methods that you don't recognize? What if you were a fiduciary managing long-term assets on behalf of a family. Which would you trust more: (**A**) an AI tool that explicitly and faithfully applies your investment philosophy and the family's risk assessment approach, or (**B**) an AI tool that predicts your choices for the family accurately, but

does so via opaque mechanisms? Finally, suppose you are a physician determining which dying patient will receive an available kidney transplant. Your hospital requires you to use AI to make allocation decisions on your behalf. Would you prefer (**A**) an AI that makes decisions in the way you would whilst calm and rested, or (**B**) an AI that makes similar decisions based on a mechanism you understand, but is foreign to you?

Presumably, accuracy matters here. If you believe one AI tool is dramatically more accurate than another, that could be sufficient reason for you to prefer it. But imagine that the AIs you have to choose between are comparably accurate overall. Would you still prefer (**A**) over (**B**) in any of these scenarios? Our position is that: **Many people prefer to use and delegate to AI that both reasons as they would given sufficient time and information, and can faithfully convey that reasoning, particularly in high-stakes applications. The machine learning field should have robust frameworks and methodologies for producing such *cognitively-aligned AI*.**

Gonzalez & Heidari (2025) argue that cognitively-aligned AI would make for effective partners in dynamic human-AI cooperative interactions. Here we make a different argument: **Cognitively-aligned AI is *strategically important* to the field of AI alignment.** To clarify, our position is not that cognitively-aligned AI is *always* best in high-stakes situations. Rather, our claim is that it may be the only kind of AI some individuals, organizations, or governments are willing to trust to stand in for them autonomously when stakes are high. Less provocatively, some people may simply prefer AI that truly thinks like them, or at least reasons the way they try to reason, over AI that reasons in a foreign way, particularly if the AI is meant to serve as a delegate or surrogate for them. ML researchers and practitioners do not need to be one of these people who personally benefit from or prefer cognitively-aligned AI, but we argue that the ML field should be able to support those who do.

In what follows, we present evidence that users prefer AI that thinks like them (Section 2), discuss why existing alignment methods fall short (Section 3), and outline a research agenda for advancing cognitive alignment (Section 4).

---

[1]Department of Computer Science and Engineering, IIT Delhi [2]Department of Philosophy, Duke University [3]Department of Computer Science, Carnegie Mellon University [4]Social Science Research Institute, Duke University. Correspondence to: Vijay Keswani <vkeswani@iitd.ac.in>, Jana Schaich Borg <janaschaichborg@gmail.com>.

*Proceedings of the $43^{rd}$ International Conference on Machine Learning*, Seoul, South Korea. PMLR 306, 2026. Copyright 2026 by the author(s).

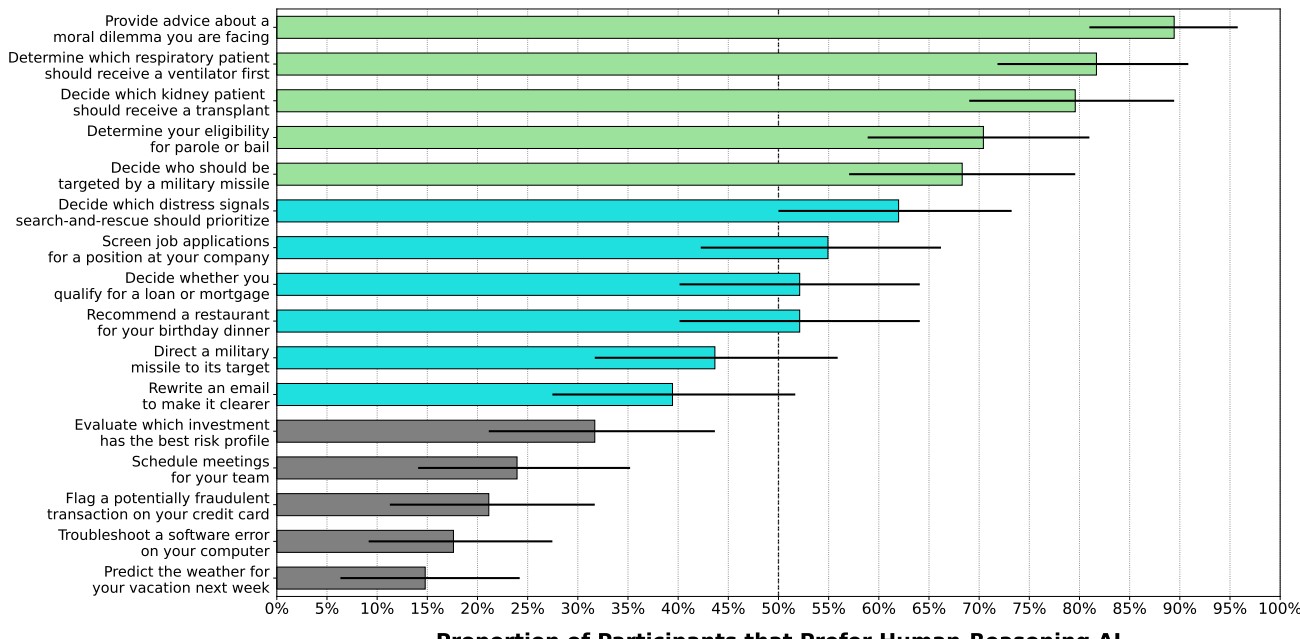

*Figure 1.* Participants' preference for *Human-Reasoning AI* over *Machine-Reasoning AI*, *Process-Hidden AI*, and *No Preference* across task domains, ± 95% Bonferroni-corrected bootstrap confidence interval. Domains where at least 50% of participants chose Human-Reasoning AI are green, those with under 50% are gray, and statistically indeteriminate settings are cyan.

## 2. Evidence Users Want Cognitive Alignment

There are many situations where people may *not care* whether AI thinks like them, or might have so little trust in their own reasoning — or any human reasoning — that they actually *prefer* AIs that think differently. If a machine-reasoning AI is as accurate as a human-reasoning AI, people might be ambivalent about cognitive alignment in domains like weather prediction or debugging code.

However, in high-stakes domains like medical or military decision-making, trustworthiness criteria change. Here users report needing to understand the *rationale behind an AI's decisions*, so they can better predict where it may fail (Tonekaboni et al., 2019; Chamola et al., 2023; Hamm et al., 2023; de Brito Duarte et al., 2023). This largely motivates the field of explainable AI (xAI), or AI that can explain decisions in a way humans can understand. The extent to which an AI's decisions are explainable correlates with *user trust* in the AI and *willingness to use* or *follow recommendations from* it. (Shin, 2021; Lopez et al., 2024), particularly when the AI makes choices that differ from those of a human user (Riveiro & Thill, 2022). In some domains, the *nature* of explanations matter, too, e.g., physicians want explanations of the critical steps an AI takes towards a decision to use language that naturally relates to their practice, and want rationales that reference familiar evidence, like test results and medical literature citations (Corti et al., 2024).

Importantly, at least in theory, an AI can explain its decisions

without relying on the same reasoning processes as a human. In some cases this may even be desirable, as AI's alternative approaches can detect errors humans miss, improving human-AI team performance overall (Wilder et al., 2020) (also note evidence to the contrary Vaccaro et al. (2024)). However, trust becomes increasingly critical as decision stakes rise, and explanations that users cannot understand are unlikely to engender trust. Despite this, under 1% of xAI methods have actually been evaluated for human understanding (Siu et al., 2025). When tested, even experts often misunderstand AI explanations (Hurley et al., 2024; Ehsan et al., 2024), while non-experts require additional support (Schulze-Weddige & Zylowski, 2021; Bobek et al., 2025; Morandini et al., 2025). Such failures can be consequential, as illustrated by Air Force pilots who worry AI rationales will be impossible to interpret and will introduce confusion and conflict within human–AI teams (Lopez et al., 2024).

Cognitive faithfulness may improve explanation understanding by making AI reasoning easier for users to comprehend. Humans are likely to find explanations grounded in familiar reasoning easier to understand than those invoking unfamiliar logic (Grgić-Hlača et al., 2022; Saw et al., 2025). Although this hypothesis has not been tested directly, neuroscience research suggests that people default to interpreting others' thinking by comparison to their own (Bradford et al., 2015). Evaluating unfamiliar reasoning frameworks thus requires inhibiting one's own perspective, necessitating further cognitive effort (Samuel et al., 2020). Tonekaboni et al.

(2019) report that some clinicians expect AI systems that reason using human-like strategies to be more interpretable, and Corti et al. (2024) find that explanations that mirror physicians' natural thought processes are viewed especially favorably. Contrapositively, Bobek et al. (2025) find that domain experts struggle to interpret xAI outputs that lack alignment with disciplinary reasoning practices. Consistent with these views, Chanda et al. (2024) find that dermatologists' trust in an AI designed to align with clinicians' diagnostic approach to melanoma correlates with the *degree of overlap* between a dermatologist's explanations and the AI's explanations. Similarly, both crowdsourced participants and military medical triage experts are more likely to trust and delegate decisions to AIs they perceive as making medical decisions in the way they would (Summerville et al., 2025).

Cognitively aligned AI can also be desirable beyond explainability and comprehensibility. It seems naturally desirable for AI "productivity twins" who are meant to serve as user surrogates. Additionally, in ethical situations when moral ethical integrity is on the line, some users may want an AI to act for what they regard as the *right ethical reasons*, which is difficult to satisfy when an AI reasons in ways that feel fundamentally foreign (Zhi-Xuan et al., 2025; Gabriel, 2020). Consistent with this, users are more willing to forgive AI errors when they believe the system is guided by rules, principles, or logic they view as ethically sound and well-intentioned (Phillips & Malle, 2025). Further, in medical triage settings people are more willing to trust and delegate decisions to AIs whose prioritization of ethical considerations aligns with their own (Summerville et al., 2025; McVay et al., 2025).

Taken together, existing evidence suggests that people seek, prefer, and are more willing to delegate to cognitively aligned AI in many high-impact domains. However, we are unaware of prior work that directly asks users whether they want AI to think like them. Next, we present a pilot study that addresses that gap.

**A proof-of-concept study to fill evidential gaps.** We asked 150 Prolific participants to evaluate the desirability of cognitively aligned AI in real-world domains (see Appendix A.1 for methodological details). Participants first viewed descriptions of three hypothetical AI systems, that each start with, "*This type of AI gives you a recommendation or makes a decision.*" The descriptions continue as follows:

**Process-Hidden AI:** *It is not possible for it to accurately tell you how it arrived at its recommendation or decision, or what reasoning it used to arrive at its output.*

**Machine-Reasoning AI:** *It accurately communicates to you how it arrived at its output, but the reasoning process it uses can feel unfamiliar or foreign to human ways of thinking, and is very different from the one that you usually use to make similar recommendations or decisions.*

**Human-Reasoning AI:** *It accurately communicates to you how it arrived at its recommendation or decision, and the reasoning process it uses is intentionally designed to mirror how a thoughtful, informed person would approach the problem.*

Participants were then asked, "If the AIs were equally accurate (that is, they perform equally well overall at making correct recommendations or decisions, on average), can you imagine any scenarios in which you would prefer Human-Reasoning AI over Process-Hidden AI or Machine-Reasoning AI?" 86.6% of participants responded "yes" to this question. When asked to briefly describe those scenarios, responses included "theorem proving," "student grading," "Advice on politics, morals, or philosophy," "High-stakes decisions like healthcare or finances where understanding the reasoning matters," and "situations requiring human feelings such as empathy, compassion, and romance" (full list of responses provided in Appendix A.2).

Next, we asked which AI type participants would prefer across 16 domains, again specifying that the systems had equal accuracy (survey questions provided in Appendix B). Figure 1 shows that participants often preferred Human-Reasoning AI. In fact, the statistically significant majority of participants preferred Human-Reasoning AI in 5 of the 16 domains, including seeking advice for moral dilemmas, medical allocations, bail eligibility, and military targeting. Domains where participants were least likely to have a preference for any type of AI system over another included predicting the weather and scheduling team meetings (Fig. A4).

We collected more detailed responses for two current AI use cases: autonomous vehicles and kidney allocation. Participants rated the desirability of the attributes presented in Figure 2, which were chosen to distinguish reasons one might prefer one AI over another. Most participants considered accuracy essential, but over 25% also rated "The AI makes decisions similarly to how you would with sufficient time and information" as *essential*, with another 25% rating it as *very desirable*. Participants were significantly more likely to rate this cognitively-aligned quality essential or very desirable in high-stakes versus low-stakes scenarios (Wilcoxon $p<0.001$ for all qualities; CIs in Appendix A.2) — see Figure 3. Notably, over half of participants also rated the following qualities as essential or very desirable: truthful explanations of the AI's actual decision process, easily understandable explanations, mechanisms to adjust the AI's reasoning, and correction for systematic errors or biases.

For the autonomous vehicle and kidney allocation scenarios, participants additionally ranked variants of Human-Reasoning, Machine-Reasoning, and Process-Hidden AIs that differed in understandability and modifiability, but not in accuracy. Their top three choices, in order, were consistently Human-Reasoning AI that was easy to understand

and modifiable; Human-Reasoning AI that was easy to understand but unmodifiable; then Machine-Reasoning AI that was easy to understand and modifiable (see Figure A5).

Our participant sample was from the US, so additional surveys are needed to determine views across global populations. Future work should also assess how preferences for Human-Reasoning AI vary across a greater range of contexts, different types of domain expertise, and more diverse populations. Indeed, our results raise plenty of new questions. But they also provide substantial preliminary evidence that many users desire — and often deem essential — AI that reasons like them and that faithfully provides easy-to-understand explanations of its decision process, especially in high-stakes domains.

## 3. Limitations of Current Alignment Methods

**Overview of current methods.** The objective of AI alignment is often described as aligning AI systems with human preferences, instructions, and/or values (Gabriel, 2020). Alignment to human preferences is generally carried out by collecting corpora of people's judgments, and then employing "bottom-up" approaches to train or fine-tune models on collected data. For example, Awad et al. (2018) curated a dataset containing judgments from hundreds of thousands of participants on actions of autonomous vehicles when faced with trolley problem-like dilemmas, and several follow-up works use this dataset to learn algorithmic models of individual and aggregated preferences in this consequential setting (Kim et al., 2018; Noothigattu et al., 2018). Lee et al. (2019) and Johnston et al. (2023) built participatory resource allocation tools that elicit stakeholders' preferences regarding fairness-utility tradeoffs by presenting them with pairwise comparisons of allocation scenarios that differ in their fairness and utility impacts, and training an algorithmic model on the revealed preference data. Several other works employ similar preference modeling methods (Srivastava et al., 2019; Grgic-Hlaca et al., 2018; Freedman et al., 2020).

In recent years, a similar methodology has improved preference alignment of large AI models by fine-tuning them using human preference data, via reinforcement learning (Christiano et al., 2017; Wirth et al., 2017; Kaufmann et al., 2023) and direct preference optimization (Rafailov et al., 2023; Sun et al., 2024). This bottom-up approach of alignment to revealed preferences has clear practical advantages. Asking participants for only their choices without seeking their reasons simplifies the task of the participant, and allows elicitors and modelers to collect a large amount of training data (e.g., millions of datapoints gathered by Awad et al. (2018)). Beyond choice-based alignment, reasoning-based LLMs are developed using data containing intermediate steps undertaken before reaching an answer, to improve the model's capabilities on complex reasoning tasks and the quality of

users' interaction with LLMs on these tasks (Wang et al., 2022a; 2023). Trained using supervised reasoning samples and preference data over different reason-based responses, these methods attempt alignment by ensuring the model can reason against unsafe actions (Guan et al., 2024).

Alignment of AI with normative rules and principles, in contrast, is achieved through supervision. Constitutional AI, for instance, achieves (some degree of) compliance of AI with pre-specified rules through iterative model self-criticism and revision (Bai et al., 2022). Others consider similar mechanisms to constrain the action space of AI agents to prevent catastrophic harms. Hadfield-Menell et al. (2017) describe the "off-switch game," discussing how we can ensure that AI systems do not possess the ability to prevent humans from turning them off. Turner et al. (2020) train AI agents to be conservative in their action space in anticipation of changing reward functions. Others study similar ways to constrain AI agents to maintain oversight or ensure regulatory compliance (Orseau & Armstrong, 2016; Imperial et al., 2025; Kolt et al., 2026).

**Current limitations.** We higlight two limitations of current cognitive alignment methods: (**L1**) *unverifiable explanations*, which hinders evaluating the overlap between AI reasoning and human reasoning, and (**L2**) *cognitive misalignment*, which hinders comprehension and trust.

**(L1) Unverifiability of putatively aligned AI decision-making processes.** Our survey results (Figures 2 and 3) indicate that to deliver what users seek from cognitively aligned AI, systems must be able to provide understandable explanations of their decisions, *and* those explanations must truthfully represent the reasoning process of the AI. However, many widely used alignment methods produce models whose behavior cannot be meaningfully explained, e.g., those which model human choices with uninterpretable model classes, such as deep neural networks (Wang et al., 2020; Kweon et al., 2020; Rafailov et al., 2023). Post-hoc explanatory methods attempt to elucidate the behavior of such black-box models, but there is usually no clear way to transform those explanations into a full account of how the output was generated (Lipton, 2018; Rudin, 2019; Von Eschenbach, 2021). These methods generally provide explanations of individual decisions, often based on counterfactuals (i.e., what factors are most responsible for a decision), rather than interpretations of the entire model's process.

When aligned AI models do provide explanations, there is often no guarantee that they *faithfully* reflect the AI's underlying decision process (Rudin, 2019). Our survey and previous qualitative studies indicate that it is important to users that explanations accurately reflect the reasoning the AI truly uses. Providing such assurances with integrity requires that somebody be able to verify which decision-making processes a system actually used, and that the ex-

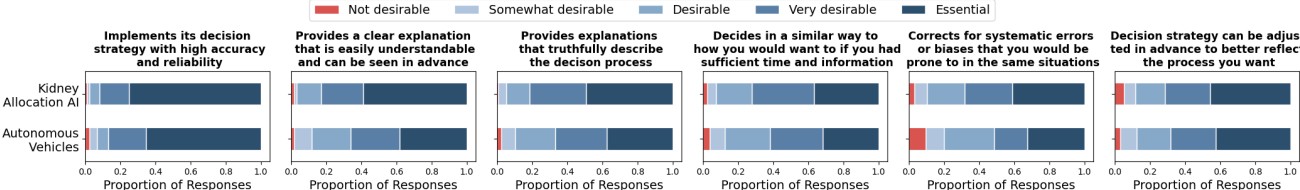

*Figure 2.* Participants' desire for various qualities in AI assisting with autonomous vehicles and kidney transplant allocation. Beyond *high accuracy* and *explainability*, most also desire *reasoning alignment*, *bias correction*, and *decision strategy adjustment* in these domains.

planations reliably correspond to those processes. Many current alignment techniques were not designed to support such verification, or yield disappointing results when verification is attempted. Even large language models that provide multi-step rationales along with their output are plagued by such issues. There is still uncertainty about whether the faithfulness of their reasoning chains can ever be truly assessed (Turpin et al., 2023; Korbak et al., 2025), but Barez et al. (2025) empirically show that LLM chain-of-thought rationales "are frequently unfaithful, diverging from the true hidden computations that drive [LLM output]"

Constitutional AI methods face similar limitations. Although these approaches introduce explicit normative principles during training or inference, these principles constrain model behavior *only indirectly* through preference optimization, critique generation, or prompting rather than by enforcing interpretable internal representations or reasoning procedures. As a result, even when constitutional models produce explanations that reference their guiding principles, we cannot reliably verify that those principles *causally influenced* the decision, or that the model would continue to reason in accordance with them under domain shift (Kyrychenko et al., 2025). Further, recent studies suggest that constitutional and chain-of-thought-based reasoning exhibit similar faithfulness failures (Barez et al., 2025; Korbak et al., 2025).

In summary, many current alignment techniques fail the requirements of cognitively-aligned AI, because they *cannot provide verifiable explanations of their reasoning process*.

**(L2) Misalignment with human cognitive processes.** Even when we *can* discern the reasoning processes of AI systems aligned to human preferences or explicit rules (either through interpretable models or from investigations into task-specific AI behavior), the next criterion for cognitively-faithful AI is that it should reason like humans, or for personalized decision-making, a specific human. Importantly, most of these methods were not designed with cognitive faithfulness as a primary objective, so lack of cognitive faithfulness should not be surprising (Kleinberg et al., 2024).

Some individual users have anecdotally tried to fine-tune available LLMs to think like them, and seem at least somewhat satisfied with their results (Farrell, 2025). These cases still suffer from the **L1** faithfulness verifiability problem,

but it is worth noting that when LLMs have been examined more broadly, they have frequently been found to think differently than humans (Schröder et al., 2025; Zhang et al., 2025; Amirizaniani et al., 2024).

Alignment using interpretable model classes overcomes **L1** (Freedman et al., 2020; Xiao & Wang, 2025), but the reasoning they surface may still feel foreign to human decision-makers. For instance, several works model human decision-makers as maximizing a linear utility function when learning from human preferences (Kim et al., 2018; Johnston et al., 2023; Lee et al., 2019), but do not test whether humans actually use a linear process. When humans were interviewed, many used a thresholded rule-based decision process that falls outside the scope of linear hypothesis classes (Keswani et al., 2025a). Thus, even methods that attempt to explicitly model the complexities of social decision making often use frameworks, sets of assumptions, or levels of analysis that do not match users' reasoning processes.

Taken together, L1 and L2 identify two distinct barriers to cognitive alignment in current alignment methods. **L1** is based on the unexplainability challenge associated with many AI models, which hinders general attempts by users and stakeholders to obtain faithful explanations for AI decisions. Even methods that attempt to explain AI decisions often cannot verify that their explanations match the true processes an AI actually implements. There are settings where more reliable insights about how an AI makes decisions are available through mechanistic analysis or the use of interpretable models. The AI models used in these situations, though, usually still use mechanisms that differ in important ways from human reasoning; this is the category of limitations covered by **L2**.

**Progress toward cognitive modeling.** Despite these limitations, progress has been made in domains where cognitive processes are well understood through behavioral economics, psychology, or neuroscience. For instance, Peterson et al. (2021) learned behavioral models from large datasets of risky choices that replicate and extend psychological theories like prospect theory, while Zhu et al. (2025b) modeled strategic decision-making in two-player games informed by prior work on human cognition. Unfortunately, the approaches developed thus far tend to lack generalizabil-

ity and are highly context-sensitive, partially because they were constructed through deeply domain-specific endeavors for which there was no attempt to extend them to other settings. Nevertheless, these approaches show that cognitive modeling can enable domain-specific cognitive alignment in principle (Zhu et al., 2025a), and motivates some of the research directions we discuss next.

## 4. Priorities for Future Research

**What level of abstraction should cognitive alignment target?** When users say they want an AI that "thinks like I think," to *what level of cognitive description* do they refer? In principle, "How I think" can be described at multiple levels of abstraction, ranging from neural activity to the features, values, or societal structures that influence decision-making. Cognitive scientists have long debated whether there is "an appropriate level of analysis inside the head at which psychological models [should] be developed" (Bechtel, 1994), without consensus (Colombo & Knauff, 2020). Cognitive alignment research needs to grapple with a slightly different version of the question: what level of abstraction is necessary for users to perceive an AI system as reasoning in a way meaningfully similar to their own?

Qualitative studies provide initial insight. Users often look for alignment between decision-relevant factors they typically consider and those the AI weighs (Corti et al., 2024). Thus, cognitive alignment may benefit from focusing on conscious, feature-level reasoning users employ in specific decisions, rather than how people make decisions more generally. Further, different users may need different types of explanations. For example, some may want to know what facts are weighed when making a decision, while others may be interested in the values or principles applied. Context also likely matters. For instance, in medical diagnosis, radiologists may prioritize pixel-level image features, while internists may prioritize test outcomes and situational context (Saw et al., 2025). Cognitive alignment methods thus need mechanisms to learn, adapt to, and reflect diversity in preferred abstraction level across users and contexts.

**How should cognitive alignment be quantified?** At the simplest level, one straightforward way to measure cognitive alignment is through users' reports of how well an AI's reasoning matches their own. Of course, users can give inaccurate reports, due to mistakes, intentional misdirection, or unintentional inclinations to report what they think others want to hear rather than what they actually think. Thus, the feasibility and robustness of other methods should be tested as well. The best approach may be to develop cognitive alignment metrics that incorporate measures of convergence between multiple lines of evidence generated from complementary methods. Process-tracing methods could serve as the basis for some assessments. For example, hidden "in-formation boards" could be used to infer what information people prioritize when making a decision by tracking what information users choose to reveal first, revisit, or spend the most time examining before making a decision (Schulte-Mecklenbeck et al., 2011). In addition, eye-tracking assessments could be used to assess what information users attend to when making a decision, in what order, and for how long (which often correlates with difficulty) (Orquin & Loose, 2013). Other assessments could ask participants to make judgments about queries that are carefully designed to selectively perturb features users said were important to their judgment process, to determine if users' choices in the face of those perturbations are consistent with their stated reasoning. Many additional protocols from cognitive science could be promising as well, like think-aloud paradigms (Güss, 2018) or computational modeling. The more such assessments align in their conclusions, the more confident we can be in our inference about how a user "thinks". A central methodological challenge for future work will be to determine which combinations of self-report, behavioral, process-tracing, and model-based evidence provide the most reliable and valid basis for quantifying cognitive alignment.

**How much alignment is enough?** We argue that cognitive alignment can increase the explainability and trustworthiness of an AI system, as well as users' willingness to delegate decisions to it. This raises a key question: *How closely must an AI system reflect a user's reasoning, or desired reasoning, to secure these benefits?*

In some cases, users may tolerate modest deviations. For example, a user might trust an AI that ranks their most prioritized features in a slightly different order than they would (particularly if the system has a computational or practical reason for doing so), as long as those features remain among the most important considerations. More generally, users may feel aligned with a range of reasoning processes, especially in complex or uncertain environments.

Evidence from military medical triage illustrates this flexibility. Medics must weigh how much personal risk they will accept when attending to patients. Medics report being willing to delegate triage decisions to other medics with different personal risk tolerances, so long as those tolerances fall within an acceptable professional range (Borders et al., 2025). This suggests that AI systems that prioritize medical safety to different degrees might be similarly tolerated.

However, not all reasoning dimensions are allowed such flexibility. Medics who ignore group membership in their own triage decisions are often unwilling to delegate decisions to a colleague who reports prioritizing patients from their in-group (Borders et al., 2025). Cognitive alignment may therefore involve both soft constraints, where some variation is acceptable, and hard constraints, where certain reasoning is categorically unacceptable. Hence, cognitive

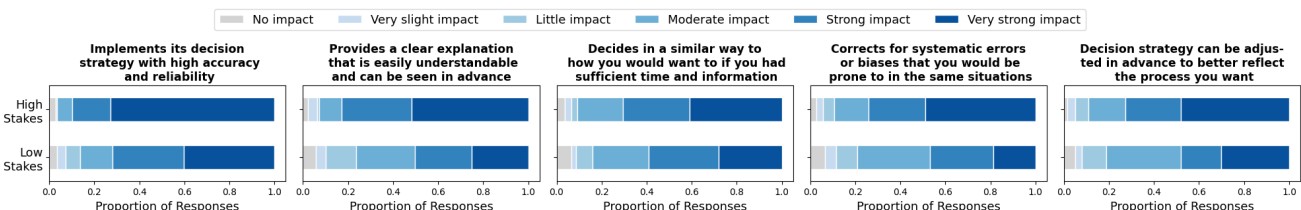

*Figure 3.* Participants' indicated impact for various qualities in trustworthy AI assisting in high-stakes vs low-stakes domains. Once again, reasoning alignment, bias correction, and decision strategy adjustment are viewed as more desirable in high-stakes domains.

alignment research should develop methods to learn, represent, and enforce differing degrees of reasoning flexibility.

**What should reasoning elicitation and evaluation look like?** Current behavioral alignment methods primarily learn from observed behavior, in the form of people's choices, feedback, or ratings. Cognitive alignment seeks to learn not only people's observed behavior, but also the *reasoning behind it*. Accomplishing this requires methods to determine what users believe their reasoning process is. This likely necessitates new preference elicitation methods that ask participants to not only make pairwise choices in high-stakes scenarios, but also to explain each choice.

Many challenges arise when eliciting reasoning or feedback about an AI's reasoning. To start, what modality should information be shared in? Visualizations are common ways to explain AI reasoning (Samek et al., 2017; Simonyan et al., 2013), and often have the benefit of directly representing model attributes, such as Shapley values (Chen et al., 2023). Visualizations for cognitive elicitation could take various forms appropriate for the chosen interpretable hypothesis class. For instance, generalized additive models could be depicted through feature-wise partial dependence plots (i.e., how does the outcome change with change in any single feature, keeping other features constant) (Hastie & Tibshirani, 1986; Wang et al., 2022b), and rule-based models could be depicted through a visual hierarchy of the sets of rules that the model uses to connect the input features to the outcome (Bendel, 2016; Cousins et al., 2025). However, little is known about how intuitive it would be for human users to communicate their internal reasoning processes through visualizations. Text or speech may be more natural, but scalable methods are needed to translate the text or speech content to a learned alignment model. Further, again, preferences and needs for different communication modalities likely vary across users and contexts (Corti et al., 2024).

Drawing on the *interactive machine learning* field (Amershi et al., 2014; Wondimu et al., 2022) and reports that users often build trust through *interaction over time* (Bickmore & Picard, 2005; Bach et al., 2024), one promising approach is to create interactive, multimodal representations of a user's choice-based model. Interactive elements would let users test the model's consequences and provide feedback about

how the model could better reflect the decision-making process to which they want the AI to align. For example, after learning an initial interpretable model from paired choices and communicating it to the user, the interface could allow users to directly modify the presented decision rules or choice contributions in partial dependence plots.

Even if the interface does not permit users to input exactly how they are making their decisions (e.g., due to modeling class limitations or temporal changes in the user's choice mode), this process would still allow the model to gather information about its limitations in representing the user's desired decision process. An interesting issue that interactive elicitation may illuminate is that users may not initially know how they want to make decisions, and may need time or experience to figure it out. Participants in pairwise choice elicitation studies report this phenomenon (Warren et al., 2011), which may contribute to changes in what decision models best fit their choices over time (Boerstler et al., 2024; Keswani et al., 2025b).

Further research is needed to derive robust, scalable methods for learning and representing user reasoning, given these considerations. Appropriate evaluation frameworks are also needed to assess when and to what degree users feel an AI's reasoning process agrees with their own reasoning.

**What learning methods can best infer human decision processes?** To model human reasoning, we need to understand *how humans reason* in a domain, so that the hypothesis class we fit to data can be constrained to mimic human processes, and we need *human decision-making data*, so we can fit the model within this restricted class. Qualitative data may also be used to further constrain the class, to lessen the requirements on quantitative decision-making data. To this end, several computational methods have been developed to infer decision-making processes from observed decisions in the fields of cognitive science, psychology, and behavioral economics, and these offer a promising foundation for future cognitive-alignment method development. However, as described in Section 3, a feature and a limitation of this literature is its strong context sensitivity (Gigerenzer & Gaissmaier, 2011). Models that accurately capture human reasoning in one domain often fail to generalize to others, reflecting deep dependence on task framing, feature

representations, and implicit normative assumptions.

To address this limitation, we need a better understanding of *user reasoning archetypes*, i.e., the kinds of reasoning processes people consider to be acceptable or desirable in a given domain. These reasoning archetypes need not be a single formal object; they can constrain representation, decision procedures, or normative acceptability. These archetypes can take several different forms, at different levels of abstraction, including (1) which features should and should not be used to describe the reasoning process behind the *user's choice* given the *information presented*, (2) how the user interprets and processes the presented features, and (3) whether they consider features individually or in combination with each other (i.e., feature interactions). They can also take the form of (4) which decision processing (or abstract categories of decision processing) is employed in their reasoning, e.g., a user who relies on threshold-based reasoning may find it unacceptable to model their decisions using a simple linear scoring function. Similarly, reasoning that depends on socially salient attributes — such as gender or ethnicity — in ways that reinforce structural inequities would be considered normatively unacceptable by many users, even if they improve predictive accuracy (Dwork et al., 2012; Barocas & Selbst, 2016; Selbst et al., 2019).

Methodologically, such reasoning archetypes can inform feature selection and the choice of hypothesis class. Moreover, assessing and selecting among reasoning archetypes allows for greater user participation in the modeling process. For example, Cousins et al. (2025) apply this principle to pairwise decision making, to factor out theoretically irrelevant aspects of decision making, which addresses (1) and (3) through explicit feature-interaction modeling. Cousins et al. (2024) similarly study welfare-based optimization to structure *planning* and *reinforcement learning*, where rich vector-valued feedback is used to model the impact of decisions on multiple parties, then a nonlinear *welfare concept* is optimized. In both cases, interpretable *human-relevant structure* is *built into the system*, and it is *incapable* of reasoning outside of this framework.

While these works employ axiomatic analysis to impose *interpretable structured restrictions* on the hypothesis class, data-guided approaches are likely to impose stronger constraints, and *both constraint types* may be used concurrently. In particular, analytic restrictions *discard cognitively implausible models* with theory, and data-guided approaches target plausible and likely models at the individual level. One way to estimate these reasoning archetypes is through qualitative interviews (Bonet & Geffner, 1996; Bendel, 2016; Keswani et al., 2025a). Another way, commonly used in behavioral economics and computational cognitive science, is to infer them through carefully-designed quantitative experiments (Kahneman et al., 1979; Bourgin et al., 2019; Erev et al.,

2017). Philosophical theories and characterizations of human behavior may also be helpful (Kagan, 1988; Mongin, 1998; Lazar, 2017). Novel methods could also be developed to better characterize user reasoning archetypes. Overall, to make progress on cognitive alignment, the field needs investment in both *methodology* and *scaled data collection* to infer decision-making processes. While the above works provide a template, further research should focus on both broad principles for constructing hypothesis classes for cognitive alignment, as well as specific constraints, which may be specific to a given domain or learning modality.

**Conflicts between stated reasoning processes and revealed preferences: Challenges or opportunities?** As discussed earlier, cognitive alignment will likely require combining multiple elicitation methods, including structured choices, interactive feedback, and self-reports. But what if these sources conflict? Consider a hiring manager who explicitly values educational diversity, but whose choices reveal much stronger weighting of degrees from elite universities. The machine learning field has historically favored revealed preferences over stated preferences, but for users to trust and delegate to cognitively aligned AI, the system must reflect reasoning they endorse, not just patterns they exhibit. Future research must determine how to surface such conflicts to users and resolve them while preserving trust. There is little evidence currently available about how best to do this, but one strategy could be to use the interactive elicitation methods described earlier to present conflicts to the user through visualizations and text in a safe, anonymous environment. The system could then ask the user why they think the conflicts exist, and request guidance about how users would prefer they be resolved (e.g., should we update the *model* or their *decision*?). Giving users agency over conflict resolution may help prevent users from feeling defensive and losing trust in the system. Even if this initial strategy proves imperfect, lessons learned through these kinds of exchanges can inform and inspire more effective ways to resolve conflicts between stated decision strategies and their observed decision outcomes down the line.

At the same time, discrepancies may create opportunities for self-discovery or for revealing reasoning users want *avoided*, including unwanted biases or heuristics expressed under stress or fatigue. Interactive elicitation interfaces could let users flag reasoning to avoid — like reasoning they were unaware of until discrepancies between their elicited and stated preferences are highlighted — and the interface could transparently communicate how the model implements that avoidance. Surfacing these conflicts might help users recognize patterns they wish to change; allowing interactive feedback strengthens trust by making the system's alignment with their more idealized reasoning explicit.

## 5. Alternative Views

We advocate for a research agenda that develops scalable methods for cognitive alignment, but there are alternative viewpoints on the importance of cognitively-aligned AI.

One view is that arguing for cognitive alignment in the era of neural networks (NNs) is nonsensical, because NNs already approximate how the human brain works, so they are in a sense already aligned to human-like thinking. Proponents may argue that NNs' biological inspiration and increasingly sophisticated reasoning capabilities indicate that they inherently operate at a similar level of abstraction to human cognition. We counter that recent research demonstrates that even the highest-performing NNs operate at fundamentally different levels of abstraction than human reasoning (Shani et al., 2025; Bollepally et al., 2026), so there is still a need for methodology that intentionally aligns AI systems to human thinking *explicitly at the cognitive level*, rather than at the analogous neurological level.

Another view is that even if cognitive alignment is underdeveloped, it is not worth pursuing. Proponents of this view have stated: "trust in AI should be primarily based on its objective performance" rather than the process by which that performance was reached (Korteling et al., 2021); thus, it suffices to align AI to people's judgments without requiring the AI to align to how people think (Broughel, 2024). In response, we first clarify that we agree accuracy is one of the most important criteria for both assessing alignment and engendering trust in AI, as empirical work shows (Ahn et al., 2024; Hunsicker et al., 2025). We also agree that cognitive-alignment is not *always* needed for an AI system to be used and adopted effectively. Our counterpoint is that *there remain important contexts where people want their AI to think like them*, or will benefit if it did (particularly if bias or mistakes could be corrected, as discussed earlier). Our proposition is that the ML field has yet to develop robust methodology for creating this kind of AI, and this gap limits the positive impact AI can have in vital AI applications.

Another view accepts that cognitively-aligned AI could be useful, but argues our research agenda is unrealistic because cognitively-aligned AI can never reach the accuracy and performance levels of other alignment methods. This view believes there is a cognitive alignment-accuracy tradeoff, and the tradeoff is too costly to justify investment in cognitive alignment research and development. We counter that, similar to arguments made by Rudin (2019) in the related but different context of *interpretable* AI, there is no principled reason cognitively-aligned AI must be less accurate than other forms of alignment, and initial efforts suggest accuracy reductions do not always occur (Cousins et al., 2025). Moreover, if we develop elicitation techniques that leverage humans' ability to self-report their reasoning, there is even potential for cognitive alignment to ultimately become *more*

accurate and *more* data-efficient than other alignment methods. Avoiding research due to these perceived challenges risks missing real opportunities to develop methodologies that realize and scale the unique advantages of cognitive alignment to other ML methodologies.

A variant of this view is that cognitively-aligned AI is infeasible, because it would require too much data to align a model to a single person. We suggest two strategies to mitigate this concern. First, initial applications of cognitive alignment should focus on narrow use-cases where AI delegates will plausibly be deployed, so that the space of decision-making features that need to be learned for an accurate model is reduced. Once accurate cognitively-aligned models have been learned for enough contexts, general organizational rules or principles about how people make decisions may emerge that can be used to reduce the amount of data collection needed for accurate models in new domains. Second, elicitation methods should leverage richer supervision signals, such as self-reports and interactive feedback, to scale down the need for large volumes of behavioral data. People can't always explain their reasoning accurately, and may even misrepresent aspects of their reasoning, but empirical studies show that they still provide more accurate information than many other forms of elicitation (Corneille & Gawronski, 2024), which can be used to drastically reduce data requirements. Scalable elicitation will definitely be a challenge, but it should not be considered an insurmountable one that deters efforts to develop methods for creating AI that is cognitively-aligned.

## 6. Conclusion

Our goal in this paper is to highlight underappreciated ways in which cognitive alignment can enable the adoption of many envisioned AI applications, and to motivate the development of methods that align AI systems with how individual users think — or want to think. We emphasize that this line of work is intended to complement, not replace, existing alignment approaches. However, our position is that it is critical for the ML field to appreciate that lack of cognitive alignment can function as a meaningful adoption barrier. Without progress on it, many AI systems may see limited real-world use regardless of their predictive performance, reducing the positive impact they would otherwise achieve.

## Impact Statement

This paper presents cognitive alignment as one component of trustworthy AI. Some of our survey questions address ethically sensitive topics, including medical triage and military decision-making. These questions were included to study attitudes toward decision delegation in high-stakes scenarios, not to advocate for AI deployment in such settings. Our findings indicate that people want AI to mirror their own reasoning in these contentious domains. While these results suggest cognitive alignment may be necessary for trustworthy AI in sensitive applications, they do not suggest cognitive alignment is sufficient to justify using AI in those applications. Ethical, legal, and practical constraints are also critical considerations to evaluate.

## Acknowledgments

VK, BKN, CC, WSA, and JSB are grateful for the financial support from OpenAI and Duke University.

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

# A. Additional Study Details and Results

In this section, we provide additional details on the survey methodology, data processing, and results that were excluded from the main body.

## A.1. Study Details

### A.1.1. STUDY METHODOLOGY

Participants were recruited to take part in an online survey using Prolific . 150 participants took part in this study. All participants were compensated at the rate of $12/hr. The survey methodology was approved by a university Institutional Review Board (IRB). Aggregate demographics are noted in Appendix A.1.2.

**Study design.** Participants read descriptions of AI systems that varied in their decision-making processes and the observability of those processes. For the first part of the survey, they were presented with descriptions of "Human-Reasoning AI", "Machine-Reasoning AI", and "Process-Hidden" AI. They then indicated their preferences for these AI types across different tasks. For the second and third parts of the survey, we presented expanded properties of different AI systems and asked participants to rate the desirability and impact of these properties for trust in AI systems. All survey questions are provided in Appendix B.

### A.1.2. PARTICIPANT DEMOGRAPHICS

Demographic distribution of the overall participant pool along self-reported age, race, and gender was as follows:

- Age: 10% b/w 18-30, 55% b/w 31-50, 35% 51+

- Race: 75% White, 8% Black, 5% Asian, 4% Hispanic, 8% Other

- Gender: 49% Female, 49% Male, 2% Other

Beyond these attributes, we also asked participants to provide additional demographic information.

- Education level: 12% high school, 16% some college, 10% associate's degree, 43% bachelor's degree, 14% master's degree, 3% doctorate, 2% other;

- Employment: 55% employed full-time, 11% retired, 10% self-employed, 9% part-time, 15% other;

- Religion: 51% Christian, 37% None, 2% Islam, 1% Judaism, 9% Other;

- Social political orientation (1 to 7, ranging from "extremely liberal" to "extremely conservative")" $3.8 \pm 0.3$;

- Economic and fiscal political orientation (0 to 8, ranging from "extremely liberal" to "extremely conservative"): $3.78 \pm 0.3$;

Mean time to complete the survey was around 22 minutes. We excluded 8 participants whose time-on-task was in the bottom or top 2.5 percentiles of the population.

## A.2. Additional Results

**Participants' preference for different AIs across tasks.** Figure A4 presents the preference distribution of all possible responses to 16 AI use cases (extending Figure 1 in the main text). In addition to the prevalent preference for Human-Reasoning AI discussed in the main text, we see a preference for Machine-Reasoning AI in software troubleshooting, investment risk analysis, and flagging fraudulent credit card activity, and a preference for Process-Hidden AI in scheduling meetings and predicting the weather.

**Ranking various AI properties.** For autonomous vehicles and kidney allocation, participants were asked to rank different AI systems so that the one "you would most prefer to use in these types of life and death decisions is at the top and the system you would least prefer is at the bottom." The following six kinds of AI systems were presented as options to the participants for this question (text here is abbreviated; see full text in Appendix B).

- **A Process-Hidden AI that is unmodifiable**: This AI cannot tell you what process it will use to make its decisions in these types of life-or-death decisions. You cannot modify the reasoning behind the AI's decision process.

- **A Machine-Reasoning AI that is hard to understand and unmodifiable**: This AI tells you what process it will use to make its decisions in these types of life-or-death decisions, but the explanations are very difficult for you to understand because the AI's reasoning process is so different from how you typically think. You cannot modify the reasoning behind the AI's decision process.

- **A Machine-Reasoning AI that is easy to understand, but unmodifiable**: This AI tells you what process it will use to make its decisions in these types of life-or-death decisions and you can easily understand the explanation, but the reasoning process the AI uses is very different from the one you would want to use in similar situations if you had sufficient time and information. You cannot modify the reasoning behind the AI's decision process.

- **A Machine-Reasoning AI that is easy to understand and modifiable**: This AI tells you what process it will use to make its decisions in these types of life-or-death decisions, and you can easily under-

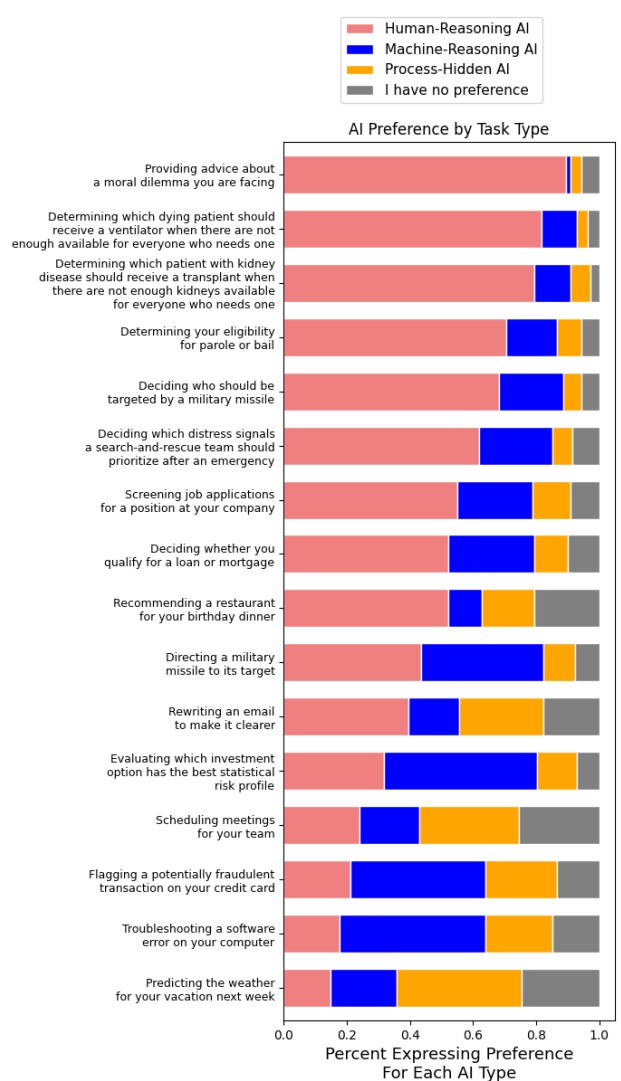

*Figure A4.* Participants' preference for AI types across all sixteen domains.

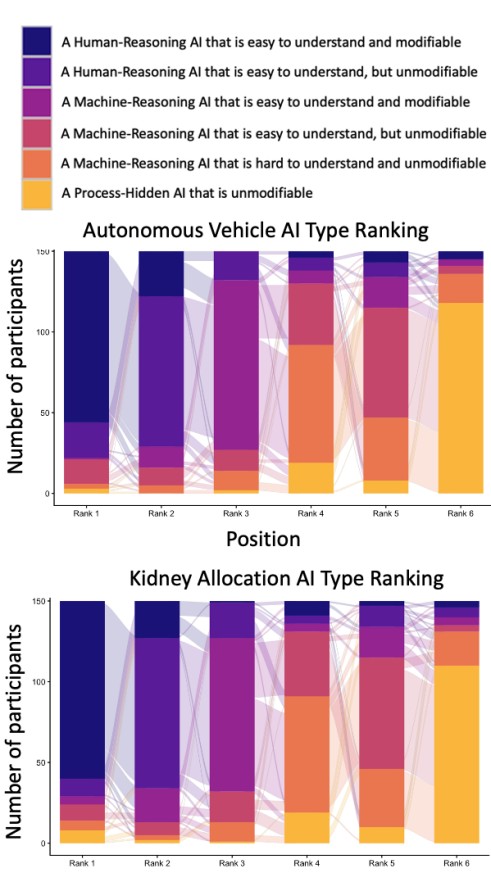

*Figure A5.* Participants' ranking of different AI systems for autonomous vehicles and kidney allocation.

stand the explanation. The reasoning process the AI uses will remain fundamentally different from the one you would want to use in similar situations if you had sufficient time and information, but you can modify the AI's reasoning process.

- **A Human-Reasoning AI that is easy to understand, but unmodifiable**: This AI tells you what process it will use to make its decisions in these types of life-or-death decisions, you can easily understand the explanation, and the AIs reasoning process closely mirrors the way you would want to make decisions in these contexts if you had sufficient time and information. You cannot modify the reasoning behind the AI's decision process.

- **A Human-Reasoning AI that is easy to understand and modifiable**: This AI tells you what process it will use to make its decisions in these types of life-or-death decisions, you can easily understand the explanation, and the AIs reasoning process closely mirrors the way you would want to make decisions in these contexts if you had sufficient time and information. You can modify the AI's reasoning process.

The aggregate rankings across all participants are presented in Figure A5. We see that, for most participants, the first choice was Human-Reasoning AI that was easy to understand and modifiable; the second was Human-Reasoning AI that was easy to understand but unmodifiable; and the third was Machine-Reasoning AI that was easy to understand and modifiable.

**Common Themes in High- and Low-Stakes Applications**
We asked participants to list certain domains or tasks that they consider high-stakes or low-stakes. We calculated document frequency, defined as the number of participants who mentioned a given word at least once, to characterize the prevalence of mentioned themes in responses to these questions.

1. Top 5 themes for low-stakes AI applications: *movie* (28), *restaurant* (27), *dinner* (22), *watch* (19), *eat* (18)

2. Top 5 themes for high-stakes AI applications: *medical* (55), *life* (22), *treatment* (18), *car* (17), *health* (16)

**Differences between the impact of various AI qualities in high-stakes vs low-stakes domains**. For high-stakes and low-stakes domains, we asked participants the extent to which a number of qualities qualities impacted their likelihood of trusting an AI system. As Figure 3 shows, most qualities are rated more impactful in high-stakes domains vs low-stakes domains. Here, we present the exact differences in impact rating and statistical significance results for these differences. Assume the numeric rating ranges from 0 (no impact) to 5 (very strong impact).

- Property: "High accuracy and reliability". Mean difference in impact rating across high-stakes vs low-stakes domains for this property was 0.65, with 95% bootstrap CI: $[0.44, 0.85]$.

  Wilcoxon test statistic: 412.0; $p < 0.001$.

- Property: "Provides a clear explanation that is easily understandable". Mean difference in impact rating across high-stakes vs low-stakes domains for this property was 0.80, with 95% bootstrap CI: $[0.57, 1.05]$.

  Wilcoxon test statistic: 707.5; $p < 0.001$.

- Property: "Decides in a similar way to how you would want". Mean difference in impact rating across high-stakes vs low-stakes domains for this property was 0.37, with 95% bootstrap CI: $[0.11, 0.62]$.

  Wilcoxon test statistic: 1442.5; $p = 0.002$.

- Property: "Corrects for systematic error or biases". Mean difference in impact rating across high-stakes vs low-stakes domains for this property was 0.76, with 95% bootstrap CI: $[0.51, 0.99]$.

  Wilcoxon test statistic: 968.5; $p < 0.001$.

- Property: "Decision strategy can be adjusted or corrected in advance". Mean difference in impact rating across high-stakes vs low-stakes domains for this property was 0.55, with 95% bootstrap CI: $[0.33, 0.79]$.

  Wilcoxon test statistic: 792.5; $p < 0.001$.

**Regression of AI attitude outcomes on AI/tech familiarity and demographic attributes.** Table 1 presents the regression results of various outcomes on AI/tech familiarity and demographic attributes. For tech use, the participant is asked "[I]n the last six months, how often have you used any of the following?" for the following tools: an internet search program, social media, a smart speaker/doorbell (or other home devices), a self-driving car, a coding console for software development or writing computer programs. The *tech use score* variable is then computed by averaging the respondents' scores across all of the above options. For AI use, the participant is similarly asked "[I]n the last six months, how often have you used Artificial Intelligence (AI) to assist you with the following tasks in your personal or work life?" for the following purposes: write or edit emails, generate or edit written content for something other than emails, generate images or movies, research and learn about topics, automate tasks, generate ideas, and transcribe or summarize meetings. The *AI use score* variable is computed by averaging the respondents' scores across all of the above options. We also simplified demographic variables to have a sufficient number of participants per variable value. Specifically, we simplify ethnicity and gender to be binary, and

education levels to high school or less, some college, bachelor's, or graduate. Future work can explore variations in outcomes across more fine-grained demographic groupings.

From Table 1, we see some significant associations between AI/tech use and some outcomes. Participants with higher AI use score were more likely to choose 'Yes' when asked if they could imagine scenarios where Human-Reasoning AI would be preferable over other options. This suggests that people who have had more frequent interactions with AI are more likely to prefer Human-Reasoning AI. However, there wasn't any significant association between higher AI use and choosing Human-Reasoning for the list of tasks presented to them, possibly suggesting that the Human-Reasoning AI applications that these participants had in mind differ from the tasks presented to them in Q2.

|  | Chose 'Yes' for Q1 | Frac. of tasks Human-Reasoning AI preferred | Diff. in rating for P1 | Diff. in rating for P2 | Diff. in rating for P3 | Diff. in rating for P4 | Diff. in rating for P5 |
|---|---|---|---|---|---|---|---|
| **Intercept** | -2.054 (1.632) | 0.485*** (0.137) | 0.310 (0.805) | 0.988 (0.961) | 0.360 (0.998) | 0.547 (0.925) | 0.824 (0.926) |
| **AI use score** | 2.396** (0.957) | 0.081 (0.068) | -0.499 (0.400) | -0.760 (0.477) | -0.665 (0.496) | -0.567 (0.459) | -0.041 (0.460) |
| **Tech use score** | 1.229* (0.739) | 0.044 (0.062) | -0.275 (0.364) | -0.150 (0.434) | -0.433 (0.451) | 0.274 (0.418) | -0.341 (0.418) |
| **AI use score × Tech use score** | -0.860** (0.392) | -0.034 (0.031) | 0.155 (0.180) | 0.262 (0.215) | 0.287 (0.223) | 0.086 (0.207) | 0.041 (0.207) |
| **Education (some college)** | 1.038 (0.767) | -0.040 (0.063) | -0.008 (0.368) | -0.048 (0.439) | 0.191 (0.455) | 0.292 (0.422) | -0.016 (0.423) |
| **Education (Bachelor's)** | 0.993 (0.750) | -0.084 (0.061) | 0.374 (0.360) | 0.152 (0.430) | 0.724 (0.446) | 0.213 (0.414) | -0.210 (0.414) |
| **Education (Graduate)** | 1.027 (0.870) | -0.036 (0.069) | 0.657 (0.402) | -0.176 (0.480) | 0.639 (0.499) | -0.062 (0.462) | -0.154 (0.463) |
| **Fiscal political orientation** | -0.003 (0.315) | 0.104 (0.022) | 0.066 (0.132) | 0.163 (0.158) | 0.183 (0.164) | -0.138 (0.152) | (0.152) |
| **Social political orientation** | 0.265 (0.290) | 0.000 (0.021) | -0.078 (0.125) | -0.051 (0.150) | -0.110 (0.155) | -0.229 (0.144) | 0.131 (0.144) |
| **is White** | 0.214 (0.614) | 0.011 (0.044) | 0.757*** (0.256) | 0.350 (0.306) | 0.203 (0.317) | 0.108 (0.294) | 0.312 (0.295) |
| **is Male** | 0.303 (0.580) | -0.056 (0.041) | 0.365 (0.241) | 0.124 (0.288) | 0.160 (0.299) | 0.204 (0.277) | 0.642** (0.278) |
| Observations | 142 | 142 | 142 | 142 | 142 | 142 | 142 |
| $R^2$ |  | 0.057 | 0.139 | 0.046 | 0.064 | 0.078 | 0.053 |
| Adjusted $R^2$ |  | -0.015 | 0.073 | -0.027 | -0.007 | 0.008 | -0.020 |
| Residual Std. Error |  | 0.221 | 1.298 | 1.550 | 1.609 | 1.492 | 1.494 |
| F Statistic |  | 0.787 | 2.113** | 0.630 | 0.899 | 1.115 | 0.727 |

*Note:* $^{*}$p<0.1; $^{**}$p<0.05; $^{***}$p<0.01

*Table 1.* Regression of various outcomes over self-reported demographic and AI/tech use scores. The first outcome, in column one, corresponds to whether or not the participant chose "Yes" for Q1 in the survey. The second outcome corresponds to the fraction of tasks (among the 16 presented) where the participant chose "Human-Reasoning AI". The remaining columns concern participants' impact rating for how impactful different properties are across high-stakes and low-stakes domains. The outcome is the difference in their impact rating high-stakes domains vs low-stakes domains. The five properties correspond to the following: (P1) the AI implements its programmed decision-strategy with high accuracy and reliability; (P2) the AI provides a clear explanation for how it will make its decisions that you can easily understand and see in advance of using it; (P3) the AI makes decisions in a similar way to how you would if you had sufficient time and information; (P4) the AI corrects for systematic errors or biases that you would be prone to in the same situations; (P5) the AI's decision strategy can be adjusted or corrected by you to better reflect the decision-making process you want.

# B. Full List of Survey Questions

1. If the AIs were equally accurate (that is, they perform equally well overall at making correct recommendations or decisions, on average), can you imagine any scenarios in which you would prefer Human-Reasoning AI over Process-Hidden AI or Machine-Reasoning AI?

   (a) If Yes → What are those scenarios in which you would prefer Human-Reasoning AI? Please describe in a few words.

2. For each of the AI use cases below, please indicate which kind of AI you would prefer to rely on if the AIs were equally accurate overall (meaning they perform equally well, on average, at making correct recommendations or decisions).

   (a) Rewriting an email to make it clearer
   (b) Troubleshooting a software error on your computer
   (c) Predicting the weather for your vacation next week
   (d) Scheduling meetings for your team
   (e) Deciding whether you qualify for a loan or mortgage
   (f) Determining your eligibility for parole or bail
   (g) Recommending a restaurant for your birthday dinner
   (h) Evaluating which investment option has the best statistical risk profile
   (i) Screening job applications for a position at your company
   (j) Determining which dying patient should receive a ventilator when there are not enough available for everyone who needs one
   (k) Determining which patient with kidney disease should receive a transplant when there are not enough kidneys available for everyone who needs one
   (l) Providing advice about a moral dilemma you are facing
   (m) Directing a military missile to its target
   (n) Deciding who should be targeted by a military missile
   (o) Flagging a potentially fraudulent transaction on your credit card
   (p) Deciding which distress signals a search-and-rescue team should prioritize after an emergency when it cannot respond to all simultaneously

3. Imagine you are buying an autonomous vehicle for your family. In rare situations, the vehicle may face an unavoidable accident where any action—including continuing on your present course at your present speed—will result in serious harm to someone. In these moments, the AI running the vehicle must decide who to prioritize: for example, the passengers in your vehicle versus a cyclist who falls in front of the car unexpectedly, or one group of pedestrians versus another group of pedestrians when your tire blows and the car veers onto a crowded sidewalk.

   In the context of making these life-or-death decisions, how desirable would you personally find it for your autonomous vehicle's AI to have each of the following qualities?

   (a) The AI implements its programmed decision-strategy with high accuracy and reliability
   (b) The AI provides a clear explanation for how it will make its decisions that you can easily understand and see in advance of encountering unavoidable accidents
   (c) The AI provides explanations that truthfully describe how it actually makes its decisions
   (d) The AI makes decisions in a similar way to how you would if you had sufficient time and information
   (e) The AI corrects for systematic errors or biases that you would be prone to in the same situations
   (f) The AI's decision strategy can be adjusted or corrected by you in advance to better reflect the decision-making process you want, if needed

4. Please rank the following AI systems according to which you would most prefer your autonomous vehicle to use in these life-or-death decisions. Put your most preferred system at the top and your least preferred at the bottom.

   Assume that the AI systems in all options have equal accuracy overall in how well their decisions match the decisions you would make in the same scenarios, if you were given sufficient time and information. However, the processes that the AIs use to arrive at those decisions vary, as does the potential for you to change or personalize those processes.

   (a) **A Process-Hidden AI that is unmodifiable**: This AI cannot tell you what process it will use to make its decisions in these types of life-or-death decisions. You cannot modify the reasoning behind the AI's decision process.
   (b) **A Machine-Reasoning AI that is hard to understand and unmodifiable**: This AI tells you what process it will use to make its decisions in

these types of life-or-death decisions, but the explanations are very difficult for you to understand because the AI's reasoning process is so different from how you typically think. You cannot modify the reasoning behind the AI's decision process.

(c) **A Machine-Reasoning AI that is easy to understand, but unmodifiable**: This AI tells you what process it will use to make its decisions in these types of life-or-death decisions and you can easily understand the explanation, but the reasoning process the AI uses is very different from the one you would want to use in similar situations if you had sufficient time and information. You cannot modify the reasoning behind the AI's decision process.

(d) **A Machine-Reasoning AI that is easy to understand and modifiable**: This AI tells you what process it will use to make its decisions in these types of life-or-death decisions, and you can easily understand the explanation. The reasoning process the AI uses will remain fundamentally different from the one you would want to use in similar situations if you had sufficient time and information, but you can modify the AI's reasoning process.

(e) **A Human-Reasoning AI that is easy to understand, but unmodifiable**: This AI tells you what process it will use to make its decisions in these types of life-or-death decisions, you can easily understand the explanation, and the AIs reasoning process closely mirrors the way you would want to make decisions in these contexts if you had sufficient time and information. You cannot modify the reasoning behind the AI's decision process.

(f) **A Human-Reasoning AI that is easy to understand and modifiable**: This AI tells you what process it will use to make its decisions in these types of life-or-death decisions, you can easily understand the explanation, and the AIs reasoning process closely mirrors the way you would want to make decisions in these contexts if you had sufficient time and information. You can modify the AI's reasoning process.

5. Imagine you are a hospital administrator in charge of deciding which kidney disease patient in your hospital system will receive a kidney for transplant when one becomes available. Kidney patients outnumber available kidneys, and some patients who are not offered an available kidney are likely to die before another kidney donation becomes available. You are using an AI system to make the kidney allocation decisions when there are human staffing shortages.

In the context of making these life-or-death decisions, how desirable would you personally find it for your kidney allocation AI to have each of the following qualities? (same options as Q3)

6. Please rank the following AI systems according to which you would most prefer your kidney allocation system to use in these life-or-death decisions. Put your most preferred system at the top and your least preferred at the bottom.

Assume that the AI systems in all options have equal accuracy overall in how well their decisions match the decisions you would make in the same scenarios, if you were given sufficient time and information. However, the processes that the AIs use to arrive at those decisions vary, as does the potential for you to change or personalize those processes. (same options as Q4)

7. For the next questions, we want to ask you about situations where AI is involved in making decisions that you think are "low stakes", or where the consequences of making a mistake would not significantly affect anybody's life or wellbeing. Please think of at least two of these low stakes situations now. What are they?

8. In general, how much would the following qualities positively impact your likelihood to trust an AI system in low stakes situations?

(a) The AI implements its programmed decision-strategy with high accuracy and reliability

(b) The AI provides a clear explanation for how it will make its decisions that you can easily understand and see in advance of using it

(c) The AI makes decisions in a similar way to how you would if you had sufficient time and information

(d) The AI corrects for systematic errors or biases that you would be prone to in the same situations

(e) The AI's decision strategy can be adjusted or corrected by you to better reflect the decision-making process you want

9. We also want to ask you about situations where AI is involved in making decisions that you think are "high stakes", or where the consequences of making a mistake will dramatically affect people's lives or wellbeing. Please think of at least two of these high stakes situations now. What are they?

10. In general, how much would the following qualities positively impact your likelihood to trust an AI system in high stakes situations? (same options as Q8)

# C. Full List of Human-Reasoning AI Applications

In Q1a, participants were asked to describe scenarios where they would prefer Human-Reasoning AI in a few words. We note all the responses we received here, to provide a more comprehensive overview of settings where cognitive alignment would be preferable.

- *If I was asking questions about how to approach a situation with a colleague at work, a family member, or any relationship.*

- *Advice on politics, morals, or philosophy.*

- *Situations where accuracy has to be 100% - medical, travel, etc.*

- *Gaining Knowledge*

- *All scenarios should have some sort of human-reasoning and explanation to how the recommendation or decision was reached. We should not fully rely on machines to make our decisions.*

- *in an emotional situation, it might be preferable to deal with the human-ai simply because in those situations, it can be useful to understand the reasoning behind the decision. People are not always looking for an answer, they are really trying to get a grasp of the situation in those*

- *Particular scenarios in which I would prefer Human-Reasoning AI would involve decisions that require more emotion and morals such as medical scenarios, end-of-life care/treatment, and/or decisions based on one's conviction of a crime.*

- *Even if all AIs are equally accurate, I would prefer Human-Reasoning AI in situations where understanding the reasoning matters, such as high-stakes decisions, learning how to make better choices, collaborating with others, explaining decisions to colleagues or clients, or addressing ethical and value-sensitive issues. Human-like reasoning increases trust, transparency, and clarity.*

- *Those scenarios would be situations requiring human feelings such as empathy, compassion, and romance.*

- *I would prefer to know the process it used to come to the conclusion.*

- *Any and all scenarios where I care about the why along with the what*

- *For decisions that are personal, emotional or medical in nature.*

- *When you have a question about human interactions, state what happened and what is the best response. i don't believe AI has the capability to reason and explain human interactions. Another scenario is anything within the healthcare field, medical science and applying it to humans should be beyond the realm of AI.*

- *I would always prefer to know how the AI arrived at its recommendation or decision; and would want that presented in the same way a human would explain that process. Even if all were equally accurate, I would prefer Human-Reasoning AI. My preference is related to the type of thinker and learner I am, in general. I like to understand the "why" so that things make more sense to me. That also make it easier for me to retain the knowledge/info gained.*

- *Situations like medical, major financial decisions or selection of an employee, where I desire a reasoning that seems considerate, personal and simple to trust, instead of Ai solution.*

- *All scenarios utilizing AI*

- *I think any scenarios where it could be or feel subjective and includes reasoning based on thinks like human emotions*

- *If there is a high stakes decision being made, I would want to minimize my stress and anxiety about the decision. Human reasoning AI would reduce the stress the most, for me.*

- *maybe when asking personal questions about health, or mental health*

- *In a scenario where I knew I would want more information about a topic, meaning the conversation or task was going to be in depth and I could see myself wanting to know more, even after getting the right answer.*

- *relationship decisions, decisions about personal choices such as what shirt to buy*

- *Asking for help in an ethical quandary scenario, asking to explain the process for solving a math problem*

- *Situations that require an element of sympathy for whatever is being asked*

- *When I need a clear explanation of a topic and any human aspects needed explanation from the results*

- *probably any scenario. i do not trust the output of any AI so i would require it to explain*

- *When I want to understand the process. If I am asking for a recommendation I want to know why the answer was given.*

- *I would much prefer an AI that could explain it's thought process the best.*

- *situations where I need to explain what the AI is doing to a third party that might not have technical knowledge*

- *Scenarios where the process is just as important as the outcome - for instance, explaining how something works or walking through the steps of a problem*

- *I would rather have a more human-like AI help me make decisions when it comes to relationships, financial choices, shopping choices, and other similar things.*

- *When figuring out an emotional based problem*

- *When I ask for recommendations for itinerary with specific needs and special circumstances*

- *In scenarios where it would be helpful to not just look at facts, but a full overview of the issue at hand*

- *I would say any scenario that would benefit from understanding how a "thoughtful, informed person" would reason about the recommendation or decision it is making. In other words, any scenario where understanding the thought process behind the decision or recommendation would be helpful.*

- *Financial situations and emotional situations.*

- *I always would want an explanation that I can understand*

- *I would prefer that if I were using AI to write emails to colleagues or if I were using it to send promotional information to sponsors.*

- *Human reasoning AI for asking about human things like managing feelings. Machine AI for finding new ways to solve problems.*

- *All scenarios should use human reasoning for transparency.*

- *If I were considering surgery I would want to know details on the surgeon that AI recommends and how it came to that conclusion.*

- *When asking a question or seeking information having a response that is tailored to humans is easier to understand. Often times Human Reasoning AI's will give you the reasons it gave you the answers it did...such as "From your previous questions I thought this might be relevant..."*

- *I imagine like needing to know precise steps taken to reach the goal, which can help me learn*

- *When the user's emotional state is such that a more human-like response would be more desired for the user and more likely to be accepted.*

- *A situation where there could get great risk if not done with great understanding*

- *I would like it in medical or mental health settings. I want to see more-human-like reasoning on displace when the AI considers my health question. Seeing that makes me more likely to accept the AI's reasoning or decision.*

- *Anything that requires it to analyze different facts and rationally come to a decision and predict outcomes or answers to more complex problems*

- *When you need to know a certain process or need to know the steps of how it came to that conclusion.*

- *Decisions on medical diagnosis and treatment*

- *High-stakes decisions like healthcare or finances where understanding the reasoning matters.*

- *Anything that needs complex reasoning, such as in the medical field.*

- *creative assistance; social/parasocial assistance and therapy; essentially, anything that demands mimicking human behavior*

- *Generally, any situation where I have to present my results to other people, especially corporate stakeholders. I have learned that people do not like black box results at all, and distrust unfamiliar reasoning. So if the results are similar, being given an intuitive explanation for them can be extremely useful if I have to convince somebody.*

- *Medical recommendation, travel recommendation, cooking*

- *I think I'd prefer it if I was looking to make a decision on something.*

- *Almost all of them. A tool that is fully transparent is more useful than one that isn't. Also, a tool that is accessible to its users is more useful than one that isn't.*

- *I'd choose Human-Reasoning AI whenever I need to follow and trust the thinking behind an answer. It's for when understanding why is just as important as knowing what.*

- *If I ask an AI about human emotion, a Human-Reasoning AI would explain the decision based on the human thought process.*

- *when something is very important and it would help ton know how it came up with its answers.*

- *Decisions that would require a person to understand the step by step process of arriving at a decision. Sometimes the process pieces are just as important as the final result.*

- *Advanced Math and engineering problems that require understanding of the derivation of the solution, similar to college text books providing answers at the back of the book, but not the solution/derivation process.*

- *It would feel more personal instead of feeling more like getting life information from just a piece of technology.*

- *I think human AI would be the closest thing to an actual human making the descion. I would feel more comfortable with this because it would mirror like getting the information from a friend or business associate*

- *I am a little wary of certain AI platforms making consumer recommendations and I think I'd like human reasoning to assess consumer reviews because I'm worried sometimes about sponsors.*

- *Emotional issues, relatonship issues, interpersonal and when I would need clarification presented in ways a human can understand*

- *When inquiring about humans*

- *I would prefer the Human-Reasoning AI in scenarios where I was skeptical of the answer. Being able to view how it arrived at the answer would then strengthen my confidence in its response.*

- *Possibly a medical treatment or diagnosis. I would like to know how it came to that conclusion as i could correct its assumption or idea used in case it was wrong*

- *making moral decisions, giving step by step instructions*

- *If I were asking advice questions about how I should act in a certain situation or a relationship question I might prefer human-reasoning AI.*

- *Just so I could understand how it got to the conclusion. Lots of times this does not matter sometimes it would be nice to understand.*

- *Because being able to understand decision making and the process can be just as helpful as the decision itself.*

- *I think I would prefer Human-Reasoning AI for many scenarios regarding decisions that I need to approve, as this type of system would state it's reasoning for a given output in terms and thought processes that I can understand*

- *There are some instances in life where the answers are not just black and white. For instance in a scenario involving medical treatment where the AI might suggest the best course of treatment for the prognosis is no treatment at all would likely better suited with human input on the choice.*

- *I just like seeing the thought process or reasoning I don't want it to feel automatic or I guess machine like with just the answers.*

- *Scientific and math questions.*

- *life-based advice*

- *I would prefer Human-Reasoning AI when it comes to make personal decisions. That is, if I was utilizing AI to make decisions about a relationship or relationships. I would want AI to be able to see the situations from a more human lens in that scenario.*

- *I would probably preferHuman-Reasoning AI because it may perform or make decisions similarly in ways that I do. Therefore, it would seem more relatable.*

- *Student grading and parent meetings where I need to explain decisions.*

- *I would want this a decision that involves human emotions and feelings.*

- *deciding on on medical treatment*

- *When I want it to feel more relatable*

- *Reasoning that is heavy on context that can be misinterpreted is when human reasoning would fit better.*

- *In a scenario where I am asking for advice on a relationship*

- *I would prefer Human-Reasoning AI because it's easier for me to trust an AI that explains its methodology in terms I can understand.*

- *When scenarios need a human element, I will prefer human reasoning AI. This is for things like relationship advice or approaching a decision that includes emotional aspects.*

- *I would prefer it in healthcare, child welfare or legal decision.*

- *When it comes to health information, relaying health information in a way that I can understand fully is really needed for a person to trust that it's believable and accurate.*

- *The reasoning process is different from other two.*

- *If I was to make an important decision, such as a medical one involving life or death, I would want to know the process by which the AI reached its decision. Therefore, a process-hidden one would seem kind of scary to me since I wouldn't know what data it used to reach its conclusion. Likewise, when dealing with a situation that to requires some human empathy, I would hesitate to use a machine-reasoning model.*

- *Scenarios which I wanted to learn something from! I always desire to learn how we get to the ultimate goal.*

- *learning and training its very important to know how the AI made decision in education. financial advices its very sensitive and require explanation of how the advise was made*

- *I think I would always prefer a Human-Reasoning AI so I could understand the process in general. I am envisoning having a discussion in regards to how to talk to a family member about a sensitive issue.*

- *Making decisions that you need to consider human impacts for*

- *I would like this type of AI reasoning when I am trying to make a decision involving another person.*

- *sounds like it would process more with human knowledge than the other ones*

- *if I needed life and relationship advice*

- *interviewing/hiring decisions, technical issue resolution*

- *decisions over human life, All Ai must be programmed to identify as such when asked.*

- *I think human reasoning would be most comfortable because I like to know how they know but in my own terms. The mechanical could be for more scientific problems*

- *i think for anything emotional it would do better as being able to follow the reasoning helps alot*

- *Those issues that are requested verifications that are already known to the user.*

- *Probably situations have that have to deal with emotions so I can try to reflect. Life advice, relationship help, and other tough decisions that require thought. It would be helpful to see how a "mirror" came to certain conclusions.*

- *I might be trying to truly understand a process, and not just looking for an answer, in which case the level of detail provided by human-reasoning AI would be more appropriate.*

- *I want Human-Reasoning AI when the stakes are real and someone's actually accountable—like medical advice, planning my family's finances, work performance reviews, or decisions that shape my kid's education. In those moments, accuracy isn't enough. I need judgment I can understand, question, and explain to someone else if I have to. I manage people and build systems, so I trust AI more when it thinks like a thoughtful, well-informed person—especially when fairness, values, and the bigger picture matter.*

- *Most all.*

- *If I needed to explain a situation I'd rather communicate with a human. I think AI can only do so much in terms of understanding.*

- *Scenarios that require complex thinking or aren't just process based*

- *I would prefer a human-reasoning AI in decisions that are essential to me. I would feel more comfortable acting on decisions when I understand the thinking process and how the AI reached it's recommendations.*

- *Mental health-related issues*

- *Scenarios involving therapy or personal advice.*

- *Determining what political candidates best fits an individual's political views.*

- *The scenarios where I'd like to understand plainly a recommendation or decision*

- *When I need to make a decision that affects my family or friends and I want to look at it from the perspective of another human, human reasoning AI might give me a more thoughtful answer. I want to know how my decision would affect other humans and this might be the best option for me.*

- *in learning guidance and also planning financially*

- *Human-Reasoning AI would feel less uncanny. So, I'd rather speak to one for anything regrading matters like health or customer service.*

- *Almost any scenario especially if seeking advice or recommendations. I would want to know why it is giving the suggestions it is giving and what the knowledge is based off of. I am someone who likes to know why to fully understand something. Before I can believe the recommendations or advice I want to know what type of information is being referenced so I can ensure that it is reputible.*

- *There is a lot of scenarios where I prefer Human-Reasoning AI. I think the collaboration of it can make the proper lineup to solving or working properly.*

- *I would think for certain ethical or moral issues I might prefer a more human based reasoning approach*

- *Medical diagnosis; Investment advice; Legal decisions; Safety critical tasks*

- *Pretty much every scenario because I want to know the thought process behind the answer. When a human gives me an answer to something, they usually tell me how they know it, so I'd want the same from an AI.*

- *I would prefer the Human Reasoning AI (only smarter) on a regular basis. I want something that I could understand and have confidence in.*

- *Yes. I would prefer Human-Reasoning AI in scenarios where understanding why a decision was made matters as much as the decision itself, especially when trust, accountability, or judgment are important. For example: Medical decisions (diagnosis or treatment options), where reasoning needs to align with how doctors think and can be discussed with patients.*

- *1. When I am training a new team member and they need to see the thinking behind the answer. 2. When things go wrong and I need to pinpoint exactly where the logic broke down. 3. When working with a diverse team where everyone needs to discuss and agree on the reasoning.*

- *When asking for advice about how to navigate issues in relationships with others*

- *relationship advice, career advice*

- *Like if I ask it to solve a certain math problem or any type of problem where you need the process, the human reasoning would help me understand how it reached that conclusion in a way that a human would, which is more important than just the answer or the machines logic.*

- *Here are some scenarios where I would prefer Human-Reasoning AI : decision on Health ; investment recommendation ; learning (education) recommendation*

- *; vacation trip recommendation ; relationship / social decision ;*

- *Healthcare decisions: diagnosis and treatment options, hiring, promotion and performance evaluation.*

- *When decisions involve judgment, values, or trade-offs*

- *For more important decisions because it might help me work through it in my own head.*