# OpenReview forum: "Position: We Need Practical AI Alignment Methods that Mirror Human Reasoning"
_ICML.cc/2026/Position_Paper_Track — ICML 2026 Position Paper Track regular_

### Official Review · Reviewer_roUZ · 2026-03-06

**Significance:** 3
**Argument Clarity:** 2
**Ethics Flag:** Yes
**Rating:** 5
**Confidence:** 5

**Questions:**

See Weakness.

**Alternative Views Section:**

Yes

**Compliance With Llm Reviewing Policy A Conservative:**

Affirmed.

**Discussion Potential:**

4

**Ethical Review Concerns:**

The survey (mainly described in section 2) included controversial scenarios, such as determining the priority order for patients receiving organ transplants or selecting targets for military strikes (e.g., missile attacks). In my view, such questions should be carefully reviewed whether they can appear in ICML, as they potentially involve human rights issues, ethical controversies, or politically sensitive topics.

**Ethics Review Area:**

["Inappropriate Potential Applications & Impact (e.g., human rights concerns)"]

**Final Justification:**

The authors have adequately addressed my concerns in the rebuttal, particularly by clarifying the distinction and relationship between L1 (evaluation difficulty) and L2 (alignment difficulty), and by explicitly acknowledging the limitations of stated preferences versus real-world behavior. I also appreciate the added discussion on ethical considerations and the commitment to clarifying the framing of sensitive scenarios. Overall, I maintain my positive assessment and recommendation for acceptance.

**Paper Summary:**

This paper argues that current AI applications has overlooked cognitive alignment (i. e., the degree to which an AI system's reasoning process matches how a chosen individual actually think or want to think, which is preferred by users), particularly in high-stakes decision-making scenarios. The authors present evidence and new survey data to support their view. Furthermore,  the authors discuss why existing alignment methods fall short in cognitive alignment, and outline a research agenda for advancing current AI product in cognitive alignment.

**Position:**

Yes

**Position In Title:**

Yes

**Related Work:**

4

**Strengths And Weaknesses:**

Strengths:

1. The position is clearly and thoroughly articulated, grounded in specific scenarios and supported by evidence. This makes the paper's argument easy to understand and persuasive.
2. The discussion of existing methods is comprehensive, and the analysis of current limitations is logically coherent.
3. The proposed solution is theoretically feasible, with the interactive design being particularly insightful and inspiring.
4. The paper addresses a timely and significant topic in the AI field,  especially concerning the practical deployment of AI technologies.

Weaknesses:

1. The distinction between L1 and L2 in the "Current limitations" section is not sufficiently clear. Since the difference in reasoning between AI and humans (L2) is a major reason why evaluating the overlap between AI and human reasoning is challenging (L1), these two points need to be more clearly delineated. While the analysis is correct, the relationship between these limitations requires clarification.
2. Regarding the interview data: As noted on page 7 in the section "Conflicts  between stated reasoning processes and revealed preferences: Challenges or opportunities?", stated preferences may not align with revealed preferences in real-world contexts. This suggests that although interviewees may express a preference for cognitively aligned AI, actual user behavior might differ. This is particularly relevant because the assumption that cognitively aligned and non-aligned AI perform equally  well often does not hold in practice (indeed, the data presented in Figures 2 and 3 indicate that accuracy remains a dominant  factor). This weakens the paper's persuasiveness. Given that high-stakes decision-making environments are difficult to simulate accurately and safely in real-world settings, this limitation is inherent to the topic.

 Suggestions:

1.  The authors may discuss the unavoidable issue of Weakness 2.
2. A few typographical errors:
   - Page 5, line 236, left column: an extraneous bracket ("[the LLM]").
   - Page 5, line 263, right column: an extraneous bracket ("[should]").
   - Page 8, line 419, right column: missed a word: "can be used to drastically data requirements."

**Support:**

3

---

> ### Author Rebuttal · Authors · 2026-03-31
>
> Thank you for the detailed and encouraging review. We are glad that you find the position well-motivated and the proposed direction meaningful.
>
> *“The distinction between L1 and L2 … need to be more clearly delineated”*
>
> Thank you for pointing this out. Our intent is:
> - L1: the difficulty of evaluating alignment between human and AI reasoning,
> - L2: the difficulty of achieving alignment due to differences in representation and abstraction between human and model reasoning.
>
> L1 is based on the unexplainability challenge associated with large models, such as those based on neural networks or ensemble methods. The unexplainability of these modeling classes hinder general attempts by users and stakeholders to obtain *faithful* explanations from AI about how it makes its decisions. Even when attempts have been made to explain AI decisions in a post hoc manner, we observe a gap between the generated explanations and the true processes that the AI actually implements.
>
> Yet, in specific applications, we may have some insights on how AI makes decisions (through mechanistic analysis or when using interpretable models). To the extent that we are able to truthfully explain AI behavior in these domains, we observe that they still differ from human reasoning. This is the category of limitations covered by L2.
>
> We will revise the text to make this distinction and their relationship more explicit.
>
> *“Regarding the interview data… stated preferences may not align with revealed preferences in real-world contexts…”*
>
> We thank the reviewer for making this point. We agree this is an inherent methodological limitation, and we now include a discussion on this point in the revised paper.
>
> That said, even if the expressed preferences for cognitively-aligned AI in our collected survey data do not fully reflect participants’ actual preference in real-world settings, their stated preference nonetheless still captures people’s idealized desire for cognitive alignment. We believe this aspiration by itself provides justification for the AI and ML community to research methods to achieve cognitive alignment in AI systems.
>
> *On ethical considerations.*
>
> We appreciate the concern regarding the use of sensitive scenarios. These were included to study decision delegation in high-stakes contexts, not to advocate for automation of such decisions. We will clarify this framing to avoid misinterpretation and add a “Broader Impacts” section discussing this point.
>
> One of our main findings is that in these ethically sensitive scenarios, real people want AI to mirror their own reasoning, thus we believe our findings actually suggest that extra caution be taken in these domains. We feel strongly that, as AI is already actively used in many of these settings, it would be irresponsible not to study them, however, we will clarify that we are not advocating for these usage, nor should these concerns be viewed as complete. We will be sure to express that cognitive alignment is only part of the picture, and if anything, these findings should be taken as necessary but not sufficient towards applying AI in sensitive domains: even achieving cognitive alignment does not override other factors in the discussions of whether AI should be used at all.
>
> *On typos.*
> Thank you for catching these; we will correct them in the revision.
>
> We appreciate the reviewer’s feedback and will incorporate these revisions to strengthen the paper.

---

> > ### Author Rebuttal · Reviewer_roUZ · 2026-04-01
> >
> > I don't have any other questions and will maintain my score.

---

### Official Review · Reviewer_smEi · 2026-03-11

**Significance:** 4
**Argument Clarity:** 3
**Rating:** 4
**Confidence:** 3

**Questions:**

1. The survey sample is demographically limited. Do the authors plan to validate the results with more diverse populations to assess generalizability?

2. The paper defines cognitive alignment broadly but does not provide a concrete metric for measuring it. How do the authors propose operationalizing cognitive alignment in future research to enable quantitative evaluation of alignment methods?

3. The research agenda mentions addressing "conflicts between stated reasoning and revealed preferences." What preliminary strategies do the authors suggest for resolving such conflicts without eroding user trust?

4. Given the dynamic nature of human reasoning, how do the authors envision cognitive alignment methods adapting to these dynamics in real-world applications?

**Alternative Views Section:**

Yes

**Compliance With Llm Reviewing Policy A Conservative:**

Affirmed.

**Discussion Potential:**

4

**Final Justification:**

Thank you for the clear response and the concrete steps proposed for improvement. I keep my original score.

**Paper Summary:**

This position paper advocates for the development of practical cognitive alignment methods in AI. It emphasizes that AI systems, especially those used in high-stakes decision-making, should reason similarly to humans and faithfully communicate their reasoning processes. The authors attempt to explore an important concept of cognitive alignment, which goes beyond mere result accuracy to focus on aligning AI’s reasoning with human cognitive patterns. The paper supports its position with a combination of existing literature reviews and original survey data. A pilot study of 150 participants shows that 86.6% prefer human-reasoning AI in specific scenarios, with a significant majority favoring it in high-stakes domains like moral dilemma advice, medical allocation, and bail eligibility. The authors aim to examine the concept by identifying two core limitations of current alignment methods and outlining a research agenda. This agenda focuses on abstraction level definition, alignment sufficiency, reasoning elicitation and evaluation, human decision process inference, and resolving conflicts between stated and revealed preferences.

**Position:**

Yes

**Position In Title:**

Yes

**Related Work:**

3

**Strengths And Weaknesses:**

Strengths:

1. This paper focuses on user trust and acceptance, which are key barriers to the real-world deployment of AI in highstakes applications. By emphasizing cognitive alignment, it occupies a valuable middle ground between traditional preference alignment and explainable AI, addressing users’ need for interpretable and verifiable reasoning.

2. The survey design compares process-hidden, machine-reasoning, and human-reasoning AI systems across a range of domains, providing concrete, user-centric evidence to support the paper’s claims.

Weaknesses:

1. The survey sample is demographically skewed and relatively small, which may limit the generalizability of results to more diverse populations or global contexts.

2. While the paper defines cognitive alignment broadly as “reasoning similarly to humans,” it provides limited concrete guidance on how to measure or quantify alignment beyond user self-reports. This lack of operationalization could hinder the translation of the research agenda into practice.

3. The research agenda acknowledges the need for scalable reasoning elicitation and evaluation but does not deeply address practical challenges or provide preliminary solutions.

**Support:**

3

---

> ### Author Rebuttal · Authors · 2026-03-31
>
> Thank you for the positive assessment and helpful suggestions. We are encouraged that you find the framing and empirical motivation valuable.
>
> *“The survey sample is demographically skewed…”*
>
> We agree that our sample is US-centric and not fully representative of global populations. We view this study as an initial, proof-of-concept investigation rather than a definitive characterization of global preferences. Even within this limited sample, we observe strong, consistent signals that many users prefer (and often require) AI systems that reason in human-like ways and provide interpretable, faithful explanations of their decision processes. How these preferences vary across contexts, particularly in domains where norms of reasoning and trust may differ, and validating our findings with domain experts and more diverse populations are important directions for future work. We will clarify this limitation and expand discussion of cross-population validation in the revision.
>
> *“paper defines cognitive alignment broadly but does not provide a concrete metric for measuring it…”*
>
> Thank you for raising the need to discuss ways to quantify cognitive alignment. We believe the problem of constructing appropriate measures and quantification for cognitive alignment is an important research direction, and will highlight the same in the paper.
>
> The current draft suggests some initial directions for assessment and operationalization in §3 and §4. We discuss obtaining users’ self-reports on cognitive alignment of AI systems (Lines 293-319, col. 1), developing novel methods to elicit users’ reasoning (Lines 278-316, col. 2), and learning from experimental procedures in behavioral economics, psychology, and cognitive science, to assess cognitive alignment (Lines 235-251, col. 2). Nevertheless, your comment has made us realize that these points are disjointly presented, and we will add a unified paragraph on quantification of cognitive alignment in the revised version.
>
> *“What preliminary strategies do the authors suggest for resolving such conflicts without eroding user trust?”*
>
> Given limited existing research on this topic within the AI alignment field, this is an important priority for future research that will require significant investment and innovation.  One strategy to start with is to provide a safe, anonymized environment where users can express their reasoning, without external pressures.  Then, using interactive elicitation methods, the system can present conflicts to the user and through visualizations and text, ask the user why the user thinks the conflicts exist, and request guidance for how they would prefer that the conflicts be resolved.  Even if challenges arise through this initial method, lessons learned through this kind of exchange can inform subsequent transparent strategies for resolving conflicts between stated decision strategies and decision outcomes.  We will expand the discussion on this point in the paper.
>
> *“Given the dynamic nature of human reasoning, how do the authors envision cognitive alignment methods adapting to these dynamics in real-world applications?”*
>
> We agree that reasoning is context-dependent and evolving, suggesting that, in practice, cognitive alignment should be adaptive and iterative, incorporating ongoing user feedback and allowing users to modify their decision strategies over time. We will clarify this point in the paper.
>
> Interactive systems (Lines 301-31, col. 2) can potentially help partially mitigate this issue: by identifying conflicts as they arise, and by not overweighing any one piece of evidence. Exposing conflicts and hypotheses to the user in the form of questions could help them identify missing contextual information, as well as to identify shifts in thinking and relearn as necessary. More specifically, reasoning descriptions and decisions can both serve as evidence. When one contradicts the other (i.e., a substantial drop in conditional probability), the system can flag the discrepancy, clarify outdated/incorrect information with the user, and update accordingly. The proposed interactive setting enables exactly this kind of iterative feedback.
>
> *“research agenda acknowledges the need for scalable reasoning elicitation…but does not…provide preliminary solutions”*
>
> We agree that scalability is a key challenge. Yet, given that cognitive alignment seems crucial for trustworthy adoption of AI tools in high-stakes domains, we need to pursue it despite this challenge. To address scalability challenges, we propose leveraging richer supervision signals than standard preference learning (e.g., structured reasoning descriptions, interactive feedback), reducing reliance on large volumes of behavioral data while improving interpretability. We outline some preliminary directions in Lines 400-423 and will clarify them further in the revision.
>
> We appreciate the reviewer’s feedback and will revise the paper to more clearly articulate these considerations.

---

> > ### Author Rebuttal · Reviewer_smEi · 2026-04-03
> >
> > Thank you for the clear response and the concrete steps proposed for improvement. I keep my original score.

---

### Official Review · Reviewer_VMqk · 2026-03-11

**Significance:** 3
**Argument Clarity:** 2
**Rating:** 4
**Confidence:** 3

**Questions:**

1. The author claims that cognitive consistency can be achieved by learning human decision-making and the reasoning process behind decision-making. However, this is similar to research on preference consistency and reasoning (thought chain) consistency. Please clarify the differences between cognitive consistency and the latter two or their combinations. The author's stated cognitive consistency can be understood as a combination of these two.

2. How to avoid inconsistencies between users' stated self-perception and their actual self-perception when aligning cognition also needs discussion.

**Alternative Views Section:**

Yes

**Compliance With Llm Reviewing Policy A Conservative:**

Affirmed.

**Discussion Potential:**

3

**Final Justification:**

The authors’ rebuttal adequately addressed my concerns and clarified the key points I raised, so I am comfortable increasing my rating to 4.

**Paper Summary:**

This paper points out that one direction for the strategic development of current AI in trustworthy fields is cognitive alignment. In short, when using AI, users may need it to make decisions in a way similar to their own thinking. It reviews relevant research and conducted a user survey of 150 people, finding that most people (especially in medical, legal, and military settings) prefer AI that demonstrates "human-like reasoning and is explainable." Finally, it points out the gaps in existing alignment methods and proposes research directions focused on "cognitive alignment" and "verifiable reasoning explanations." While the motivation is insightful, the value of the new direction proposed in the paper is still worth discussing.

**Position:**

Yes

**Position In Title:**

Yes

**Related Work:**

2

**Strengths And Weaknesses:**

**Strengths**

The motivation of this manuscript is somewhat insightful. It conducted a survey of 150 people and pointed out that a significant number of participants preferred AI with "cognitive alignment" when making decisions on certain matters.

**Major Weaknesses**

1. How to define and achieve "cognitive alignment" from a qualitative to a quantitative perspective, and how to model cognition, requires a more detailed and specific explanation from the author. In fact, this is crucial to transforming this argument from mere discussion into practical implementation and development.

2. As the author states, the authenticity of thought chains is difficult to guarantee; the same issue arises with "cognitive alignment." How to genuinely ensure that AI achieves "cognitive alignment" requires discussion at the methodological and strategic levels.

**Minor Weaknesses**

1. Regarding the main description "cognitive alignment," if it's a positional paper with a defined stance, it's recommended that the title more clearly state that cognitive alignment is needed.

**Support:**

3

---

> ### Author Rebuttal · Authors · 2026-03-31
>
> *MW1: “How to define and achieve "cognitive alignment" … [and] model cognition requires a more detailed and specific explanation…”*
>
> We very much appreciate the reviewer’s detailed questions about how to best make the approach precise. The goal of this position paper, after all, is to get researchers excited enough about the broader agenda to want to think about how exactly it should be pursued. At the same time, the impact and scope of position papers like this one would be dramatically reduced if limited to topics with finalized technical details, or if expected to cover those details alongside motivation—a scope better suited to technical tracks. We agree that a position paper should provide enough detail to show that our proposed research agenda is workable. We sought to do that, and offer Reviewer roUZ’s statement that “*the proposed solution is theoretically feasible, with the interactive design being particularly insightful and inspiring*” as evidence that we have done so. We hope that submissions to the position paper track are treated consistently regarding the expected balance between calls to action and the details provided for their implementation (see §4).
>
> *MW2: “authenticity of thought chains is difficult to guarantee; the same issue arises with "cognitive alignment.” [Ensuring] that AI achieves "cognitive alignment" requires discussion…”*
>
> Our response to MW1 applies here as well. We discuss aspects of the methods for faithful cognitive alignment in the “Priorities for Future Research” section, but balance technical detail against the breadth of topics we want ML researchers to consider. Notably, the reviewer’s comment seems primarily inspired by challenges with LLM thought chains. Our call to action extends beyond LLMs. Many of the methodological approaches we describe could currently be achieved most directly using other ML methods, some of which do not have the same challenges with faithful explanations for decisions, even if they achieve this faithful interpretability via domain-specific modeling. In section “What learning methods can best infer human decision processes?”, we discuss how learning user reasoning archetypes can help minimize this tradeoff.
>
> *Q1: “cognitive consistency …similar to research on preference consistency and reasoning (thought chain) consistency. Please clarify the differences between cognitive consistency and the latter two”*
>
> We are unclear on what kinds of “consistency” the reviewer has in mind. One possibility is that the reviewer is suggesting that researchers study how consistent people’s expressed choices are with an LLM model’s chain-of-thought (CoT) reasoning. Another possibility is that it refers to how consistent human reasoning processes are with a model’s CoT reasoning. Yet another possibility is that the question refers to how consistent an LLM’s output is with its own stated CoT reasoning. Can the reviewer clarify what they mean by “research on preference consistency and reasoning (thought chain) consistency”?
>
> Our answer will reference the following ideas. One requirement of cognitive alignment, by our view (supported by our survey data indicating that participants want AIs to truthfully describe their decision process), is that the model must be able to accurately describe how it is making decisions. Inability to achieve this is discussed as L1 in the paper, and may differentiate the cognitive alignment we are calling for from work the reviewer might be referring to. Further, cognitive alignment targets the internal structure of the decision-making process (e.g., which features are considered, how they are combined, how counterfactuals are evaluated). Thus, it is not enough that a model makes the same decisions as a human (preference agreement); they must do so in the same way. This, too, may differentiate cognitive alignment from work the reviewer might be referring to.
>
> *Q2: “How to avoid inconsistencies between users' stated self-perception and their actual self-perception..”*
>
> We interpret this comment as pointing out that users’ stated descriptions of how they make decisions may differ from how they *actually think* they make decisions, whether due to social desirability concerns or intentional misdirection of model training. Although this problem is not unique to cognitive alignment and plagues many forms of preference elicitation, we agree that it merits discussion, and will add a paragraph in the final version of the paper on this issue, describing our view that an initial research priority should be to assess how frequently it poses a problem for cognitive alignment, and in what ways. Initial assessments could be pursued through a combination of skilled interviews, comparisons of self-reported and computationally-inferred decision processes, and responses to discrepancies unearthed through these analyses. The results of this first research stage would inform what kinds of mitigation or technical strategies are needed, and their urgency.

---

> > ### Author Rebuttal · Reviewer_VMqk · 2026-04-03
> >
> > The authors’ rebuttal adequately addressed my concerns and clarified the key points I raised, so I am comfortable increasing my rating to 4.

---

### Review · Ethics_Reviewer_xqHQ · 2026-03-27

**Recommendation:** No remediation action needed

**Ethics Issue:**

Reviewer roUZ flagged a potential ethical issue related to "Inappropriate Potential Applications & Impact (e.g., human rights concerns)", referencing potentially controversial scenarios used in the Authors' survey instrument, in particular organ transplant prioritization or targets for missile strikes.

Reading through the conference's Research Ethics page, I believe this might be better triaged as about "Responsible Research Practice (e.g., IRB, documentation, research ethics, participant consent)". This calls for authors to provide evidence that they adhered to their "home institution’s procedures for obtaining IRB approval" or were eligible for an exemption.

From what I can tell, the authors have done this. In section A.1.1 ("Study Methodology") they note that "Participants were recruited to take part in an online survey using Prolific . 150 participants took part in this study. All participants were compensated at the rate of $12/hr. The survey methodology was approved by a university Institutional Review Board (IRB)." I think this is sufficient, and within the guardrails of the conference's ethical guidelines.

roUZ, thank you for the opportunity to weigh in.

---

### Decision · Program_Chairs · 2026-04-30

**Decision:**

Accept (regular)

**Comment:**

Reviewers consistently praised the paper for its clear and insightful position, clear writing, and for inclusion of a survey that provides further color to the position. While reviewers also pointed out several weaknesses (difficulty in operationalizing the proposal, small scale of survey, and so on), these have largely been addressed during the rebuttal.

Based on the positive appraisal of the reviewers, I suggest acceptance.